# In-Context Learning of Energy Functions

## Abstract

In-context learning is a powerful capability of certain machine learning models that arguably underpins the success of today's frontier AI models. However, in-context learning is critically limited to settings where the in-context distribution of interest $p_\theta^{ICL}(\boldsymbol{x}|\mathcal{D})$ can be straightforwardly expressed and/or parameterized by the model; for instance, language modeling relies on expressing the next-token distribution as a categorical distribution parameterized by the network's output logits. In this work, we present a more general form of in-context learning without such a limitation that we call *in-context learning of energy functions*. The idea is to instead learn the unconstrained and arbitrary in-context energy function $E_\theta^{ICL}(\boldsymbol{x}|\mathcal{D})$ corresponding to the in-context distribution $p_\theta^{ICL}(\boldsymbol{x}|\mathcal{D})$. To do this, we use classic ideas from energy-based modeling. We provide preliminary evidence that our method empirically works on synthetic data. Interestingly, our work contributes (to the best of our knowledge) the first example of in-context learning where the input space and output space differ from one another, suggesting that in-context learning is a more-general capability than previously realized.

## 1. Introduction

Probabilistic modeling often aims to learn and/or sample from a probability distribution. In the specific context of in-context learning, the distribution of interest is oftentimes a conditional distribution where some data $\mathcal{D}$ is provided "in-context":

$$p_\theta^{ICL}(\boldsymbol{x}|\mathcal{D}) \qquad (1)$$

For concreteness, the in-context data might be text (Brown et al., 2020), synthetic linear regression covariates and tar-

gets (Garg et al., 2022), or images and assigned classes (Chan et al., 2022). Directly learning this conditional distribution can be straightforward if the probability distribution can be easily parameterized; for instance, next-token prediction can be readily specified as a classification problem, where the conditional distribution is a categorical distribution parameterized by the model's output logits. However, this limits the expressivity of in-context learning to situations where the conditional distribution can be straightforwardly parameterized.

In this work, we explore a more general form of in-context learning with no such constraint on how readily the conditional distribution can be specified. We call this more general form *in-context learning of energy functions*. The key insight is that rather than dealing with the constrained conditional distribution, we instead re-express it in its Boltzmann distribution form (Bishop & Nasrabadi, 2006):

$$p_\theta^{ICL}(\boldsymbol{x}|\mathcal{D}) = \frac{\exp\left(-E_\theta^{ICL}(\boldsymbol{x}|\mathcal{D})\right)}{Z_\theta}, \qquad (2)$$

where $Z(\theta) \overset{\text{def}}{=} \int_{\boldsymbol{x} \in \mathcal{X}} \exp(-E(\boldsymbol{x}))\,d\boldsymbol{x}$. This alternative form is preferable because the energy function is an arbitrary unconstrained function $E : \mathcal{X} \times \mathcal{D} \to \mathbb{R}$ that can be used to express any probability distribution without requiring a particular form. We then propose learning the in-context energy function $E_\theta^{ICL}(\boldsymbol{x}|\mathcal{D})$ rather than the constrained in-context conditional distribution $p_\theta^{ICL}(\boldsymbol{x}|\mathcal{D})$, which we accomplish by drawing upon well-established ideas in probabilistic modeling called energy-based models (Hinton, 2002; Mordatch, 2018; Du & Mordatch, 2019; Du et al., 2020).

## 2. In-Context Learning of Energy Functions

### 2.1. Learning In-Context Energy Functions

Our goal is to learn the in-context energy function:

$$E_\theta^{ICL}(\boldsymbol{x}|\mathcal{D}) \qquad (3)$$

What concretely does this mean? We seek a model with parameters $\theta$ that accepts as input a dataset $\mathcal{D}$ with arbitrary cardinality and a single datum $\boldsymbol{x}$, and adaptively changes its output energy function $E_\theta^{ICL}(\boldsymbol{x}|\mathcal{D})$ based on the input dataset $\mathcal{D}$ without changing its parameters $\theta$.

---
[1]Anonymous Institution, Anonymous City, Anonymous Region, Anonymous Country. Correspondence to: Anonymous Author <anon.email@domain.com>.

Preliminary work. Under review by the 1st In-context Learning Workshop at the International Conference on Machine Learning (ICML). Do not distribute.

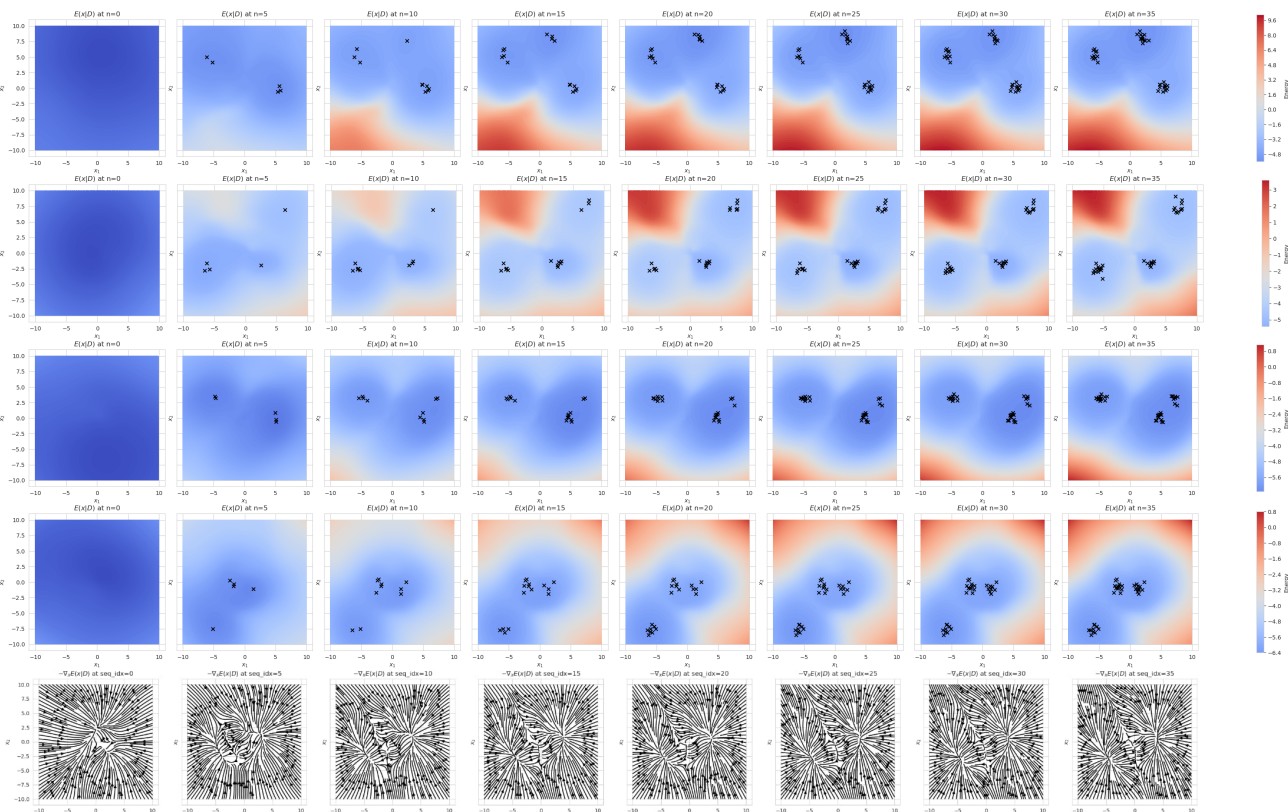

*Figure 1.* **In-Context Learning of Energy Functions.** Transformers learn to compute energy functions $E_\theta^{ICL}(\boldsymbol{x}|\mathcal{D})$ corresponding to probability distributions $p^{ICL}(\boldsymbol{x}|\mathcal{D})$, where $\mathcal{D}$ are in-context datasets that vary during pretraining. At inference time, when conditioned on a new in-context dataset, the transformer computes a new energy function using fixed network parameters $\theta$. The transformers' energy landscapes progressively sharpen as additional in-context training data are conditioned upon (left to right). **Bottom.** The energy function $E_\theta^{ICL}(\boldsymbol{x}|\mathcal{D})$ can be used to compute a gradient with respect to $\boldsymbol{x}$ that enables sampling higher probability points, without requiring a restricted parametric form for the corresponding conditional probability distribution $p_\theta^{ICL}(\boldsymbol{x}|\mathcal{D})$.

For concreteness, in the context of conditional probabilistic modeling, a causal transformer is typically trained to output a conditional probability distribution at every index, i.e.,

$$p_\theta^{ICL}(\boldsymbol{x}_2|\boldsymbol{x}_1), p_\theta^{ICL}(\boldsymbol{x}_3|\boldsymbol{x}_2, \boldsymbol{x}_1), \dots$$

Instead of learning each conditional distribution $p_\theta(\boldsymbol{x}_n|\boldsymbol{x}_{<n})$, we instead learn the corresponding energy function $E_\theta(\boldsymbol{x}_n|\boldsymbol{x}_{<n})$. This means that the transformer instead outputs a *scalar* at every index, *regardless of the shape of the inputs*:

$$E_\theta^{ICL}(\boldsymbol{x}_2|\boldsymbol{x}_1), E_\theta^{ICL}(\boldsymbol{x}_3|\boldsymbol{x}_2, \boldsymbol{x}_1), \dots$$

This scalar at each index is the model's estimate of the *energy* at the last ($n^{\text{th}}$) input datum, based on an energy function constructed from the previous $n-1$ datapoints.

To achieve this practically, we use causal GPT-style transformers (Vaswani et al., 2017; Radford et al., 2018; 2019). Just like with standard in-context learning of language models, we train our transformers by minimizing the negative log

conditional probability, averaging over possible in-context datasets:

$$\mathcal{L}(\theta) \stackrel{\text{def}}{=} \mathbb{E}_{p_{data}}\left[\mathbb{E}_{\boldsymbol{x},\mathcal{D}\sim p_{data}}\left[-\log p_\theta^{ICL}(\boldsymbol{x}|\mathcal{D})\right]\right]. \quad (4)$$

Due to the intractable partition function in Eqn. 4, we minimize the loss using contrastive divergence (Hinton, 2002). Letting $\boldsymbol{x}^+$ denote real training data and $\boldsymbol{x}^-$ denote confabulated (i.e. synthetic) data sampled from the learned energy function, the gradient of the loss function can be reexpressed in a more manageable form:

$$\nabla_\theta \mathcal{L}(\theta) = \nabla_\theta \mathbb{E}_{p_{data}}\left[\mathbb{E}_{\boldsymbol{x}^+\mathcal{D}\sim p_{data}}\left[-\log p_\theta(\boldsymbol{x}|\mathcal{D})\right]\right]$$

$$= \mathbb{E}_{p_{data}}\left[\mathbb{E}_{\boldsymbol{x}^+|\mathcal{D}\sim p_{data}}\left[\nabla_\theta E_\theta^{ICL}(\boldsymbol{x}^+, \mathcal{D})\right]\right]$$

$$- \mathbb{E}_{p_{data}}\left[\mathbb{E}_{\mathcal{D}\sim p_{data}}\left[\mathbb{E}_{\boldsymbol{x}^-\sim p_\theta^{ICL}(\boldsymbol{x}|\mathcal{D})}\left[\nabla_\theta E_\theta^{ICL}(\boldsymbol{x}^-|\mathcal{D})\right]\right]\right].$$

```
function training_step(batch):
    # Compute energy on real data.
    real_data = batch["real_data"]
    energy_on_real_data = transformer_ebm.forward(real_data)

    # Sample new confabulated data using Langevin MCMC.
    initial_sampled_data = batch["initial_sampled_data"]
    confab_data = sample_data_with_langevin_mcmc(real_data, initial_sampled_data)

    # Compute energy on sampled confabulatory data.
    energy_on_sampled_data = zeros(...)
    for seq_idx in range(max_seq_len):
        for conf_idx in range(n_confabulated_samples):
            real_data_up_to_seq_idx = clone(real_data[:, :seq_idx+1, :])
            real_data_up_to_seq_idx[:, -1, :] = sampled_data[:, conf_idx, seq_idx, :]
            energy_on_confab_data = transformer_ebm.forward(real_data_up_to_seq_idx)
            energy_on_sampled_data[:, conf_idx, seq_idx, :] += energy_on_confab_data[:, -1, :

    # Compute difference in energy between real and confabulatory data.
    diff_of_energy = energy_on_real_data - energy_on_sampled_data

    # Compute total loss.
    total_loss = mean(diff_of_energy)

    return total_loss
```

*Figure 2.* **Pseudocode for Training In-Context Learning of Energy Functions.**

This equation tells us that we can minimize the negative log likelihood by equivalently minimizing the energy of real data (conditioning upon the in-context data) context while simultaneously maximizing the energy of confabulated data (again conditioning upon the in-context data). Training Python pseudocode is given in Figure 2.

### 2.2. Sampling From In-Context Energy Functions

To sample from the conditional distribution $p_\theta^{ICL}(\boldsymbol{x}|\mathcal{D})$, we follow standard practice in energy-based modeling (Hinton, 2002; Du & Mordatch, 2019; Du et al., 2020): We first choose $N$ data (deterministically or stochastically) to condition on, and sample $\boldsymbol{x}_0^- \sim \mathcal{U}$ for some distribution $\mathcal{U}$ to compute the initial energy $E_\theta(\boldsymbol{x}_0^-|\mathcal{D})$. We then use Langevin dynamics to iteratively increase the probability of $\boldsymbol{x}_0^-$ by sampling with $\omega_t \sim \mathcal{N}(0, \sigma^2)$ and minimizing the energy with respect to $\boldsymbol{x}_t^-$ for $t = [T]$ steps:

$$\boldsymbol{x}_{t+1}^- \leftarrow \boldsymbol{x}_t^- - \alpha \nabla_{\boldsymbol{x}} E_\theta^{ICL}(\boldsymbol{x}_t^-|\mathcal{D}) + \omega_t. \qquad (5)$$

This in-context learning of energy functions is akin to Mordatch (2018), but rather than conditioning on a "mask" and "concepts", we instead condition on sequences of data from the same distribution and we additionally replace the all-to-all relational network with a causal transformer.

### 2.3. Preliminary Experimental Results of In-Context Learning of Energy Functions

As proof of concept, we train causal transformer-based ICL-EBMs on synthetic mixture-of-Gaussian datasets. The transformers have 6 layers, 8 heads, 128 embedding dimensions, and GeLU nonlinearities (Hendrycks & Gimpel, 2016). The transformers are pretrained on a set of randomly sampled synthetic 2-dimensional mixture of three Gaussians with uniform mixing proportions with Langevin noise scale 0.01 and 15 MCMC steps of size $\alpha = 3.16$. After pretraining, we then freeze the ICL-EBMs' parameters and measure whether the model can adapt its energy function to new in-context datasets drawn from the same distribution as the pretraining datasets. The energy landscapes of frozen ICL EBMs display clear signs of in-context learning (Fig. 1).

## 3. Discussion

To the best of our knowledge, *this is the first instance of in-context learning where the input and output spaces differ*. This stands in stark comparison with more common examples of in-context learning such as language modeling (Brown et al., 2020), linear regression (Garg et al., 2022) and image classification (Chan et al., 2022). Our results

demonstrate that transformers are more capable of different types of in-context learning than previously known, and our results demonstrate that transformers can successfully learn energy functions rather than probability distributions. Although our results are quite preliminary, we believe this is an exciting direction that can be pushed significantly further.

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
