# OpenReview forum: "In-Context Learning of Energy Functions"
_ICML.cc/2024/Workshop/ICL — ICML 2024 Workshop ICL Poster_

### Official Review · Reviewer_mFXz · 2024-05-30

**Rating:** 2
**Fit:** 3
**Confidence:** 3

**Workshop Review:**

This paper proposes an in-context learning of energy function, which is a classical way of parameterising some distribution without giving the normalised density.
The paper suggests the use of contrastive loss for the training of transformer models and also employs the Langevin dynamics-based sampling to sample from the density expressed by energy.
The paper also gives the preliminary experiment that shows for the class of the mixture of three two-dimensional Gaussians, the transformer can perform in-context learning of the density given the samples from this density.

## Clarity

It is not hard to reconstruct the proposed method with the text assuming that the reader has knowledge of (Garg et al. 2022), which is a base framework of this paper.

However, there is some flaw in the pseudocode (Figure 2), for example, the confab\_data is used nowhere, leading to the pseudocode not matching the proposed loss.
Also, it is not clear whether the gradient is passed through the langevin\_mcmc step in the pseudocode, which should not be.

Figure 1 is one of the main contributions in the paper, which shows that the proposed algorithm does work.
It would be better to contain the ground truth energy to compare the effectiveness of the proposed method.
Also, the Gaussian distributions used seems isotropic, and has small variance, which makes the task too simple.

There are also some minor mistakes in the equation.
(Page 2, right column, lines 102-109)
The use of $E_{x^+ | \mathcal{D} \sim p_{data}}$ and $E_{x^+ \mathcal{D} \sim p_{data}}$ is mixed, the second line should have $E_\theta^{ICL}(x^+ | \mathcal{D})$, and it is hard to see what $p_{data}$ stands for.

## Correctness

The proposed training loss is well used in the energy-based models, and also the inference algorithm is one of the most standard methods in gradient-based sampling.
There is no doubt that the proposed method is sound.

## Novelty

To the reviewer’s best knowledge, this is the first method to in-context learn the probability distribution in multi-dimensional space.
The closest work is PFN (Muller et al. 2022), which is limited to the one-dimensional space, and has a lack of expressivity by the parameterisation imposed by PFN.

## Interest

Both the density estimation by in-context learning and the integration of MCMC algorithms for in-context learning are interesting to the community.

(Muller et al. 2022) Transformers can do Bayesian Inference, Samuel Muller, Noah Hollmann, Sebastian Pineda, Josif Grabocka, Frank Hutter. ICLR 2022.

(Garg et al. 2022) What Can Transformers Learn In-Context? A Case Study of Simple Function Classes, Shivam Garg, Dimitris Tsipras, Percy Liang, Gregory Valiant, NIPS 2022.

**Reason For Not Giving Higher Score:**

While the paper proposes a novel training method to model the conditional distribution, it is yet preliminary to convince that this in-context learning would lead to a density estimation model that is beneficial compared to existing works.
It would be beneficial to the paper if there exists comparison to simple baselines like Gaussian mixture fitting, or some out-of-distribution generalisation.
Also, one of the paper’s claims, that ‘this is the first instance of in-context learning where the input and output spaces differ’, doesn’t seem true. The linear regression task (Garg et al. 2022) has input space of 20 dimensional vector and output space of single scalar, the PFN (Muller et al. 2022) has its input space as arbitrary, and output space as parameterisation of Riemann distribution.

There are several pitfalls in this paper, for example, unlike (Garg et al. 2022) or next token prediction, their training can't be performed in parallel. Also, the gradient-based sampling usually requires thousands of iteration, which becomes impractical since we require backward computation of transformer model.
For this method to give impressing result, such problems should be handled.

(Muller et al. 2022) Transformers can do Bayesian Inference, Samuel Muller, Noah Hollmann, Sebastian Pineda, Josif Grabocka, Frank Hutter. ICLR 2022.

(Garg et al. 2022) What Can Transformers Learn In-Context? A Case Study of Simple Function Classes, Shivam Garg, Dimitris Tsipras, Percy Liang, Gregory Valiant, NIPS 2022.

**Reason For Not Giving Lower Score:**

The proposed method is sound, and it opens an interesting direction to extend the in-context learning of transformers towards distribution modelling.
For example, if we believe the model selection behaviour of in-context learning (Bai et al. 2023, Yadlowsky et al. 2023) extends to this paper’s claim, we can expect the hyperparameter tuning (e.g., the number of mixtures in Gaussian mixture), model selection, and distribution modelling to happen all at once via in-context learning.

(Bai et al. 2023) Transformers as Statisticians: Provable In-Context Learning with In-Context Algorithm Selection, Yu Bai, Fan Chen, Huan Wang, Caiming Xiong, Song Mei, NIPS 2023.

(Yadlowsky et al. 2023) Pretraining Data Mixtures Enable Narrow Model Selection Capabilities in Transformer Models, Steve Yadlowsky, Lyric Doshi, Nilesh Tripuraneni, arxiv.org/abs/2311.00871.

---

### Official Review · Reviewer_Ht8p · 2024-06-08

**Rating:** 2
**Fit:** 3
**Confidence:** 2

**Workshop Review:**

This paper shows the preliminary empirical result for the idea that model can learn energy function in context instead of probability distribution. The technical approach appears sound and the training methods are based on existing techniques. The preliminary result shows that the model indeed adapting its learn energy function to new datasets. However, more extensive experiments are needed for consistency. The work explores when the input and output spaces differ in in-context learning setting, what is a good method to solve this issue.

## Clarity
The authors provide intuition for why learning in-context energy functions is a useful extension of in-context learning to more general output spaces. However, it will be better to have more clarification on pseudocode. For example, what is "sampled_data" in line 124-125?

The equation for derivative also contains some confusing terms such as $E_{x^{+}D∼pdata}$.

## Correctness
The technical approach appears sound and also used in the existing energy-based models literature. Contrastive divergence training and Langevin dynamics sampling are used.

## Novelty
This paper demonstrates in-context learning where the input and output spaces differ, moving beyond next-token prediction to learning general energy functions. This is a novel extension of the in-context learning paradigm.

## Interest to the Community

In-context learning beyond the scope of probability distribution, and the preliminary result from figure 1 shows this works

## Some questions:
1. What might happen to other architectures and more training configurations.

**Reason For Not Giving Higher Score:**

Need more experiments to show the result is consistent

**Reason For Not Giving Lower Score:**

Post an interesting perspective that the input space of in-context learning and output space of result can differ and model can actually learn this

---

### Meta-Review · Area_Chair_v6XJ · 2024-06-17

**Recommendation:** 2

**Metareview:**

Reviewers agree that the idea of in-context learning energy functions is of interest, though the empirical evaluation and broader use of the method is not yet clear. Some specific comments:
> this limits the expressivity of in-context learning to situations where the conditional distribution can be straightforwardly parameterized

Could distributions not be implicitly represented (e.g., see work on "ICL learns (parameterized) SGD"), or nonparametrically represented?

> this is the first instance of in-context learning where the input and output spaces differ

Not sure what is meant here, so it is not clear whether this point of novelty connects with any impact.

---

### Decision · Program_Chairs · 2024-06-17

**Decision:**

Accept (Poster)

**Comment:**

**Accept with minor revision**: Please address the clarity concerns of Reviewer Ht8p, include citations to more work on in-context learning of implicit algorithms (*e.g.*, [Bai et al. (2023)](https://arxiv.org/abs/2306.04637)), and use objective language when describing the contributions of your work (*cf.* "we believe this is an exciting direction").